

# Functional characterization of antennae-enriched chemosensory protein 4 in emerald ash borer, *Agrilus planipennis*

Ren Li[1], Zehua Wang[1], Fan Yang[1], Guanghang Qiao[1], Jingjing Tu[1], Ang Sun[1,2] and Shanning Wang[1]

[1] Key Laboratory of Environment Friendly Management on Fruit and Vegetable Pests in North China (Coconstructed by the Ministry and Province), Ministry of Agriculture and Rural Affairs; Institute of Plant Protection, Beijing Academy of Agriculture and Forestry, Beijing, China

[2] Fruit-Vegetable-Flower Integrated Pest Management of Invasive Pests, Yunnan International Joint Laboratory, School of Ecology and Environmental Science, Yunnan University, Kunming, China

Corresponding author
Shanning Wang,
wangshanning@yeah.net

## ABSTRACT

*Agrilus planipennis* is an invasive species that inflicts substantial harm on ash trees (*Fraxinus* spp.) globally. Elucidating its olfactory mechanisms is essential for devising effective pest management approaches. In this research, we identified chemosensory protein 4 (AplaCSP4) in *A. planipennis*, which is highly expressed in the antennae of both male and female individuals. Notably, the mRNA expression level of *AplaCSP4* in females is 1.9 times higher than that in males. Fluorescence competition binding assays revealed that recombinant AplaCSP4 has a broad binding spectrum, capable of interacting with 11 compounds from various chemical classes such as esters, alkanes, terpenes, terpenoids, and terpenols. The dissociation constants ($K_D$) for these binding affinities range from 0.25 to 11.47 µM. AplaCSP4 shows binding affinity for volatiles from *Fraxinus* species, including dodecane, myrcene, ocimene, farnesene, (+)-limonene, and nerolidol, with the highest affinity observed for farnesene ($K_D = 0.25$ µM). Molecular docking and dynamics simulation were employed to elucidate the binding mode of farnesene, which exhibited the strongest binding affinity with AplaCSP4. The results indicated that farnesene binds within the hydrophobic pocket of AplaCSP4, with a binding energy of $-31.830 \pm 2.015$ kcal/mol and $-32.585 \pm 2.011$ kcal/mol in dual-replicate molecular dynamics simulations, and primarily driven by van der Waals interactions. Importantly, during the two molecular dynamics simulations, the centroid distances between farnesene and the key residues in the binding pocket of AplaCSP4 were maintained relatively stable. The combined results from *in vitro* experiments and computational modeling suggest that AplaCSP4 is critically involved in plant volatile detection. This study offers insights into the molecular basis of olfactory perception in *A. planipennis* and may provide a foundation for developing novel olfactory-based pest control strategies targeting chemosensory proteins.

## INTRODUCTION

Insects rely heavily on their olfactory system to navigate their environment, locate food sources, find mates, and avoid predators. The olfactory system of insects is highly

specialized, with antennae serving as the primary sensory organs for detecting chemical cues. The detection of these cues is mediated by a range of proteins, including odorant-binding proteins (OBPs) and chemosensory proteins (CSPs), which play crucial roles in the initial steps of olfactory perception (*Jia et al., 2023*; *Leal, 2013*; *Pelosi et al., 2018*; *Tu et al., 2024*). Among these, CSPs are involved in the binding and transport of hydrophobic odorants through the aqueous sensillum lymph to olfactory receptors (ORs) on the dendrites of olfactory receptor neurons (ORNs) (*Hansson Bill & Stensmyr Marcus, 2011*; *Pelosi et al., 2014*; *Pelosi et al., 2018*).

CSPs are a class of small, soluble proteins that are distinct from OBPs in terms of their structure and sequence (*Brito, Moreira & Melo, 2016*; *Cheema et al., 2021*; *Wanchoo et al., 2020*). CSPs typically contain four conserved cysteine residues and are involved in a variety of functions, including chemosensation, development, immune response, and insecticide resistance (*Gaubert et al., 2020*; *Li et al., 2020*; *Li et al., 2025*; *Qu et al., 2020*; *Tomaselli et al., 2006*). CSPs are broadly expressed in insects, and their specific expression in antennae is typically linked to the recognition of volatile compounds, contributing to host recognition and mating behaviors. For example, the CSP6 and CSP7 of *Aleurocanthus spiniferus* (Quaintance) (Hemiptera: Aleyrodidae) are associated with the recognition of host plant volatiles, including (E)-2-hexenal, linalool, 3-carene, and hexanol (*Jia et al., 2024*).

The emerald ash borer, *Agrilus planipennis* Fairmaire (Coleoptera: Buprestidae), is a highly destructive invasive pest of ash trees, native to Asia but now widespread in North America and Europe (*Herms & McCullough, 2014*; *Liebhold et al., 2024*; *Morin et al., 2017*). *A. planipennis* has caused significant ecological and economic damage, leading to the death of millions of ash trees. Understanding the molecular mechanisms underlying its olfactory perception is crucial for developing effective management strategies. Although previous studies have identified several OBPs showing specific expression in the antennae of *A. planipennis*, suggesting their involvement in olfactory processes (*Andersson, Keeling & Mitchell, 2019*; *Shen et al., 2021*; *Wang et al., 2020*), the specific roles of CSPs in the olfactory system of *A. planipennis* remain poorly understood.

In this study, the functional characterization of chemosensory protein 4 in *A. planipennis* (AplaCSP4) was validated, and it showed enriched expression in the antennae of both female and male compared with other CSP genes. We aim to determine the expression profile of *AplaCSP4* in different tissues, investigate its binding affinity to various plant volatiles, and explore its potential role in the olfactory perception of *A. planipennis*. By combining molecular, biochemical, protein structure prediction, and molecular docking approaches, we seek to elucidate the role of AplaCSP4 in the detection of plant volatiles. This research will not only enhance our understanding of the olfactory mechanisms underlying plant volatiles identification in *A. planipennis* but also may provide a foundation for the development of novel olfactory-based pest management strategies.

## MATERIALS & METHODS

### Insect culture

The collection of samples referred to the method of *Wang et al. (2020)*, which is specifically described as follows. In April 2019, *Fraxinus* trees affected by the larvae of *A. planipennis* were felled in the suburban ash forests of Beijing and cut into approximately 50 cm segments. These segments were then brought indoors and placed within rearing cages. After the larvae pupated and the adults emerged, they were transferred to rearing boxes and fed with ash tree leaves for experimental purposes. The adult rearing conditions were maintained at a temperature of 25 ± 0.5 °C, a relative humidity of 60% ± 5%, and a photoperiod of 16L:8D.

### RNA isolation and first-strand cDNA synthesis

Tissues from 100 antennae, six heads (without antennae), and four bodies (a mixture of thoraces, abdomens, legs, and wings) were dissected from 1- to 3-day-old female and male beetles, separately. RNA extraction and reverse transcription were performed according to the method of *Wang et al. (2021)*, as detailed below. The tissues were immediately frozen in liquid nitrogen and stored at −80 °C until RNA isolation. Total RNA from different tissues was isolated using TRIzol reagent (Invitrogen, Carlsbad, CA, USA) according to the protocol. The integrity and purity of RNA were assessed using 1.2% agarose gel electrophoresis and a NanoPhotometer N60 (Implen, München, Germany), respectively. First-strand cDNA was synthesized from 1 μg of RNA using the PrimeScript™ RT reagent Kit with gDNA Eraser (Takara, Beijing, China), following the manufacturer's instructions.

### Sequence and multiple sequence alignment

Gene-specific primers (Table S1) for amplifying the open reading frame (ORF) of the *A. planipennis* chemosensory protein 4 (*AlpaCSP4*) were designed using the Primer 3 program (http://primer3.ut.ee/) according to the reference sequence (accession: XM_018478165.1). Cloning of the sequences was carried out following the method described by *Wang et al. (2021)*. Specifically, polymerase chain reaction (PCR) amplification was conducted using one unit of KOD DNA polymerase (Taihe, Beijing, China) and 200 ng of cDNA template. The PCR cycling parameters were as follows: initial denaturation at 94 °C for 2 min, followed by 30 cycles consisting of denaturation at 94 °C for 20 s, annealing at 58 °C for 30 s, and extension at 68 °C for 1 min. A final extension step was performed at 68 °C for 5 min. The resulting PCR products were inserted into the pClone EZ-Blunt vector (Taihe, Beijing, China). Subsequently, the cloned products were sequenced using the M13 primer for further analysis.

The amino acid sequence of AlpaCSP4 was aligned with those of CSPs from *Agrilus mali* (AXG21596.1), *Helicoverpa armigera* (AIW65103.1), *Glossina morsitans* (CBA11330.1), *Encarsia formosa* (QJT73564.1), and *Subpsaltria yangi* (AXY87875.1) using Clustal Omega (https://www.ebi.ac.uk/Tools/msa/clustalo/). The result of the multiple sequence alignment was visualized using ESPript 3.0 (https://espript.ibcp.fr/ESPript/cgi-bin/ESPript.cgi).

## Reverse transcription PCR

RT-PCR was completed according to the method of *Wang et al. (2020)*, as follows. The spatial distributions of *AplaCSP4* across the head (without antennae), body (mixtures of thoraxes, abdomens, legs, and wings), and antennae in males and females were characterized by semiquantitative RT-PCR, employing Taq DNA polymerase (Biomed, Beijing, China). Each PCR was performed in 25 μL reaction volume, containing 200 ng of cDNA from different tissues. The thermal cycling conditions for the PCR were initiated with an initial denaturation at 94 °C for 4 min, followed by 30 cycles comprising a denaturation step at 94 °C for 30 s, an annealing step at 55 °C for 30 s, and an extension step at 72 °C for 45 s. A final extension was conducted at 72 °C for 5 min to complete the amplification process. The integrity of the cDNA samples was assessed using β-actin (accession: XM_018479924.1) as a reference gene. For each amplification, negative controls with a water template were included to ensure specificity. The resulting amplification products were electrophoresed on 1.2% agarose gels to verify their size and integrity. To authenticate the identity of each gene, one amplification product per gene was subjected to sequencing. The gene-specific primers utilized in these reactions are detailed in Table S1.

## Quantitative real-time PCR

Quantitative real-time reverse transcription PCR (qRT-PCR) was performed following the protocol described by *Wang et al. (2021)*, with specific details as follows. The relative mRNA expression level of *AplaCSP4* in the antennae of males and females was measured by qRT-PCR. The qRT-PCR was conducted using the ABI Prism 7500 System (Applied Biosystems, Carlsbad, CA, USA) and SYBR Green SuperReal PreMix Plus (TianGen, Beijing, China) in a 20 μL reaction mixture. The mixture comprised 10 μL of 2 × SuperReal PreMix Plus, one μL (200 ng) of sample cDNA, 0.4 μL of 50 × ROX Reference Dye, 0.4 μL of each forward and reverse primers, and 7.8 μL of sterilized ultrapure water. Each qRT-PCR experiment was performed with three biological replicates, and each replicate was assessed in triplicate. Both β-actin and elongation factor 1-α (*EF*-1α, accession: XM_018476784.2) *EF*-1α served as endogenous controls to normalize the target gene expression and account for sample-to-sample variability. Primers for performing the qRT-PCR were listed in Table S1. The specificity of each primer set was confirmed through melting curve analysis, while the amplification efficiency was determined by evaluating the standard curves generated from a 5-fold dilution series of cDNA. The expression level of *AplaCSP4* in the antennae of female was compared relative to the antennae of male using the $2^{-\Delta\Delta CT}$ method. The AplaCSP4 gene expression level in antennae of male and female individuals was compared using *T*-test of GraphPad prime 10 software (GraphPad Software, San Diego, CA, USA).

## Expression and purification of recombinant AplaCSP4

*AplaCSP4* was amplified by PCR using specific primers (Table S1). The resulting PCR products were inserted into a T vector (Taihe, Beijing, China) and subsequently cloned into the bacterial expression vector pET30a (+) (Novagen, Madison, WI, USA), and the sequence was verified by sequencing. The plasmids of harboring the correct insert
sequence were transferred into BL21 (DE3) competent cells for subsequent protein expression. Protein expression was induced in LB medium at 18 °C for 16 h by the addition of one mM isopropyl-β-D-thiogalactopyranoside (IPTG). The bacterial cultures were harvested through centrifugation and subsequently resuspended in a 50 mM Tris–HCl buffer (pH 7.4). Following sonication and centrifugation, the recombinant proteins were predominantly present in the supernatant, and were purified by a standard Ni column (GE Healthcare, Waukesha, WI, United States). The His-tag was selectively cleaved from the recombinant proteins using a recombinant enterokinase (Novagen, Madison, WI, USA) according to the manufacturer's protocol. Purified AplaCSP4 was dialyzed in a 50 mM Tris, and the protein concentration was then accurately determined using the Bradford protein assay.

## Fluorescence competitive binding assays

Fluorescence competitive binding assays were conducted according to the method described by *Wang et al. (2021)*. Specifically, the binding affinities of AplaCSP4 for 43 volatile organic compounds were determined using F-380 fluorescence spectrophotometer (Tianjin, China) with a 10 nm slits and a 1 cm light path. These 43 substances are common green leaf volatiles, among which 16 volatiles originate from the ash tree. As the fluorescent probe, N-phenyl-1-naphthylamine (1-NPN) was excited at the wavelength of 337 nm, and emission spectra were recorded between 390 and 530 nm. Utilizing N-phenyl-1-naphthylamine (1-NPN) as the fluorescent probe, excitation was set at 337 nm, with emission spectra captured in the range from 390 nm to 530 nm. To quantify the binding affinity of 1-NPN to AplaCSP4, a 2 $\mu$M solution of the purified protein in 50 mM Tris–HCl buffer (pH 7.4) was titrated with aliquots of 1-NPN (one mM in methanol) to achieve final concentrations between 2 and 16 $\mu$M.

Competitive binding assays were conducted by titrating a solution containing both AplaCSP4 protein and 1-NPN, each at a concentration of two $\mu$M, with aliquots of a 1 mM methanol solution of the ligand, achieving final concentrations ranging from 2 to 20 mM. The dissociation constants of the competitors were determined using the equation $K_D = IC_{50}/(1 + [1\text{-NPN}]/K_{1-\text{NPN}})$, in which $IC_{50}$ represents the concentration of the ligand required to reduce the initial fluorescence intensity of 1-NPN by half, [1-NPN] denotes the free concentration of 1-NPN, and $K_{1-\text{NPN}}$ indicates the dissociation constant characterizing the AplaCSP4/1-NPN complex. The experiments were executed in triplicate for ligands that demonstrated significant binding affinity, while those ligands exhibiting minimal binding were subjected to a single experiment.

## Molecular docking and dynamic simulation

The signal peptide of AplaCSP4 was removed and subsequently used to construct a 3D protein model using AlphaFold3 (https://alphafoldserver.com/). The AplaCSP4 structure minimization was performed using the ff14sb force field on Wemol (https://wemol.wecomput.com/ui/#/). A Ramachandran plot was generated using the online tool PROCHECK to assess the quality of the constructed 3D model. Molecular docking of AplaCSP4 with farnesene was performed using CB-Dock 2 (https://cadd.labshare.cn/cb-

dock2/php/index.php). The conformation with the highest score is used for molecular dynamics simulation. The GROMACS 2024 was used to perform the protein-ligand complex molecular dynamics simulation, and the AMBER03 force field was selected in conjunction with the TIP3P water model. Energy minimization was performed using the steepest descent method (Steep), with the energy convergence threshold set at 10 kJ/mol/nm. The simulation was conducted under the NPT ensemble, with a coupling reference pressure of one bar and a temperature maintained at 300 K. The simulation time step was 0.002 ps, and the total simulation duration was 200 ns. MMPBSA was used to calculate the binding energy. With different initial velocity random assignments, the molecular dynamics simulation was repeated using the same parameters. The conformation of the last frame was used for the analysis of the interactions between the protein and the ligand by applying PyMOL 2.6.0 (https://pymol.org/) and Maestro 14.1 (https://www.schrodinger.com/).

## RESULTS

### Sequence analysis of AplaCSP4

The nucleotide sequence of *AplaCSP4* was verified by molecular cloning and sequencing. Analysis of the AplaCSP4 sequence revealed a full-length ORF consisting of 390 nucleotides that encode 130 amino acid residues (Fig. S1). The results of multiple sequences alignment displayed that the AplaCSP4 possessed the typical chemosensory protein conserved domain of C1-X6-C2-X18-C3-X2-C4 and contained seven α helices (Fig. 1).

### The expression level of AplaCSP4 in various tissues

RT-PCR was employed to assess the expression of *AplaCSP4* in the antennae, heads, and bodies of both female and male individuals. The findings revealed that *AplaCSP4* was specifically and prominently expressed in the antennae of both sexes (Fig. 2). The qRT-PCR results demonstrated that the mRNA expression level of *AplaCSP4* in female antennae was 1.91-fold higher than that in male antennae ($P = 0.0002$) (Fig. 2).

### Binding characteristic of recombinant AplaCSP4

AplaCSP4 was expressed in a bacterial system and subsequently utilized to screen for potential ligands of AplaCSP4. The protein was purified using affinity chromatography on Ni columns and employed for ligand-binding experiments. The size and purity of the recombinant protein were assessed by SDS-PAGE (Fig. 3).

The binding affinities of AplaCSP4 for 43 volatile compounds were quantified using fluorescence competition binding assays, employing 1-NPN as a fluorescent probe. The binding affinity constant ($K_D$) between 1-NPN and AplaCSP4 was determined to be 9.36 μM, thereby validating 1-NPN as a suitable fluorescent reporter (Fig. 4). The binding experiments demonstrated that AplaCSP4 exhibited binding interactions with 11 volatile compounds, encompassing esters, alkanes, terpenes, terpenoids, and terpenols, with $K_D$ values ranging from 0.25 to 11.47 μM. Among the 11 plant volatiles, six are derived from the ash tree, including dodecane, myrcene, ocimene, farnesene, (+)-limonene, and nerolidol. Notably, farnesene displayed the strongest binding affinity, with a $K_D$ value of 0.25 μM

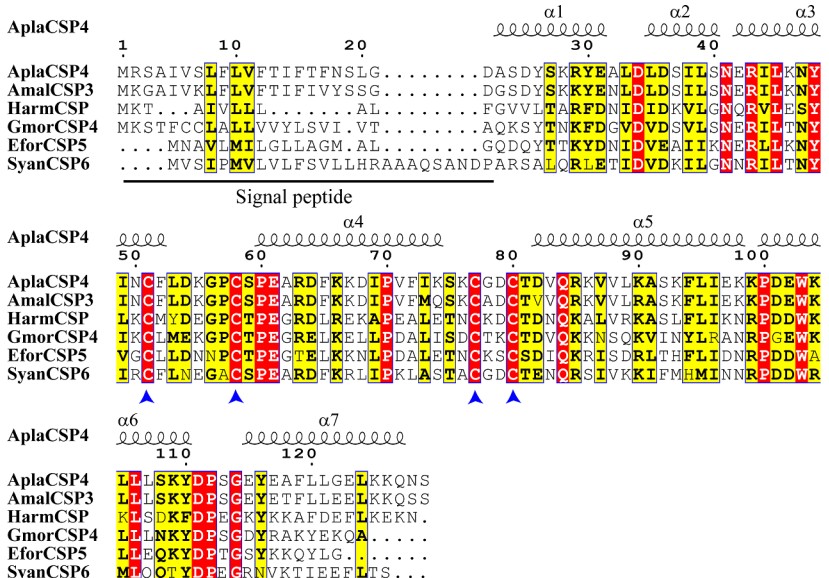

**Figure 1** **Multiple sequence alignment of *Agrilus planipennis* CSP4 (AplaCSP4) with different species.** The signal peptide is marked by a black line. The four conserved cysteines are indicated by the blue arrow. The seven α helices are represented with α1-7 according to the AplaCSP4 structure.

(Fig. 4 and Table 1). Additionally, AplaCSP4 exhibited moderate binding affinities with *cis*-3-hexenyl benzoate, *n*-octane, decane, dodecane, myrcene, nerolidol, and β-ionone, with $K_D$ values ranging from 1.38 to 9.90 µM (Fig. 4 and Table 1). Conversely, ocimene, (+)-limonene and hexyl butyrate exhibited relatively low binding affinities, with $K_D$ values of 10.43, 11.34 and 11.47 µM, respectively (Fig. 4 and Table 1).

## Molecular docking and dynamic simulation

The Ramachandran plot was used to assess the rationality of the minimized structure of AplaCSP4 and revealed that 100% of residues were in the allowed region (Fig. S2), indicating the predicted model of AplaCSP4 was reasonable and reliable. The AplaCSP4 contained 7 helices (α 1–7) (Fig. S2).

The volatile compound farnesene, which exhibited the highest binding affinity in the fluorescence competition binding assay, was selected for molecular docking with AplaCSP4. The conformation with the highest docking score was subsequently employed for molecular dynamics (MD) simulation with dual-replicate. The results of the two molecular dynamics simulations indicate that the root mean square deviation (RMSD) reaches a stable state after 125 ns (Fig. 5). Additionally, the root mean square fluctuations (RMSF) were displayed in Fig. 5. The results of the MMPBSA analysis showed that the binding energy between AplaCSP4 and farnesene was −31.830 ± 2.015 kcal/mol and −32.585 ± 2.011 kcal/mol in dual-replicate molecule dynamics simulations (Table 2). Both molecular dynamics simulations demonstrated that farnesene binds within the same hydrophobic binding pocket of AplaCSP4 (Fig. 6), with binding site coordinates at (30.558, 30.106, 31.277) and (31.075, 29.304, 31.781), respectively. The energy contributions of the amino acid residues

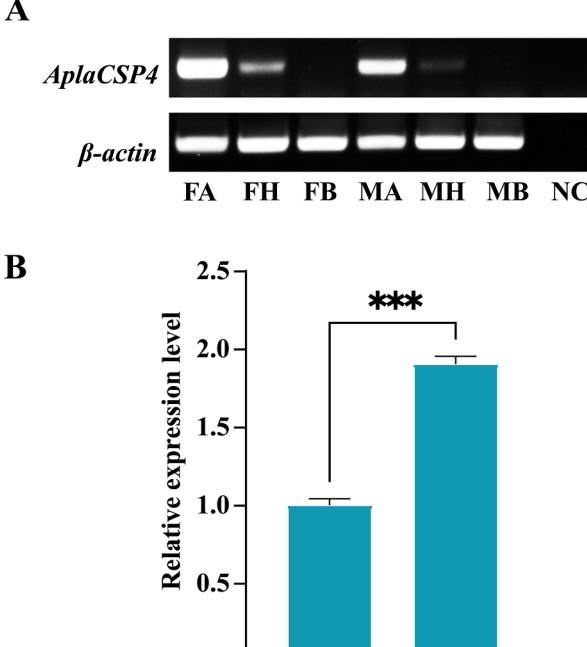

**Figure 2 Tissue-specific expression of *Agrilus planipennis* chemosensory protein 4 (AplaCSP4) gene in male and female adult.** (A) indicates the expression of *AplaCSP4* in female antennae (FA), female head (FH), female body, male antennae (MA), male head, and male body (MB). $\beta$-actin was used as a control gene. NC was negative control. (B) displays the mRNA expression level of *AplaCSP4* in FA and MA. Error bars represent the standard error (SE), and "***" indicates the *P*-value < 0.001.

in the active site are depicted in Fig. 7, where van der Waals forces represent the predominant interaction forces between the amino acids and farnesene. The centroid distance between farnesene and binding pocket of AplaCSP4 was analyzed by two molecular dynamics simulations, and the results indicated that the average centroid distance of farnesene from AplaCSP4 active center was $0.13 \pm 0.004$ nm and $0.16 \pm 0.004$ nm in both simulations, and the average centroid distance of carbon atom between farnesene and the side chain of hydrophobic amino acids was $1.41 \pm 0.003$ nm and $1.39 \pm 0.002$ nm (Fig. 8), it suggested that the farnesene has stable interaction with AplaCSP4.

## DISCUSSION

As a native pest in Northeast Asia and an invasive pest in Europe and America, *A. planipennis* has caused significant ecological and economic damage by infesting and killing ash trees in these regions (*Herms & McCullough, 2014*; *Liebhold et al., 2024*; *Morin et al., 2017*). Understanding the mechanisms underlying its host recognition and olfactory perception is crucial for developing effective pest management strategies. Some studies have reported the chemosensory proteins are critical in the transport of volatiles and pheromones in various insects, such as *Agrilus mali*, *Halyomorpha halys*, and *Frankliniella occidentalis* (*Jia et al., 2024*; *Li et al., 2021a*; *Li et al., 2022*; *Wang et al., 2021*). Previously, 14 chemosensory

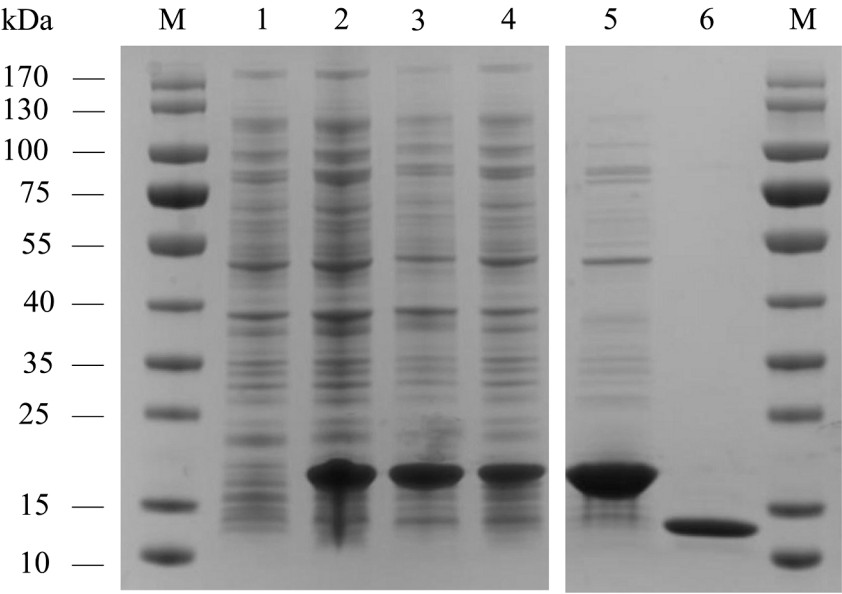

**Figure 3** **The recombinant chemosensory protein 4 of *Agrilus planipennis* (AplaCSP4) is analyzed using SDS-PAGE.** M, Protein molecular weight marker; 1, Non-induced *Escherichia coli*; 2, Induced *E. coli*; 3, Supernatant; 4, Inclusion body protein; 5, Recombinant AplaCSP4 with His-tag; 6, Recombinant AplaCSP4 without His-tag.

protein genes have been identified from *A. planipennis* genome (*Andersson, Keeling & Mitchell, 2019*). The expression profiles of these genes across various tissues were elucidated *via* RT-PCR. Notably, AplaCSP4 and AplaCSP5 were found to be highly enriched in the antennae of both female and male individuals (Fig. S1). However, AplaCSP5 failed to exhibit detectable expression in heterologous systems. AplaCSP12 (an AplaCSP4 paralog) displayed specific expression in the head (without antennae) (Fig. S3). Given these findings, this study focuses on AplaCSP4, a protein that is highly expressed in the antennae of *A. planipennis*. It shows the capacity to bind to specific volatiles tested in this study and has a binding pocket that strongly interacts with the tested ligand farnesene.

Our results showed that AplaCSP4 is prominently expressed in the antennae of both male and female *A.planipennis*. More importantly, the expression level in female antennae was 1.91 times higher than in males, indicating that AplaCSP4 might be associated with host location for oviposition behaviors in females, which aligns with previous findings that female *A. planipennis* are more responsive to host volatiles than males (*Rodriguez-Saona et al., 2006*). This sex-specific expression pattern highlights the importance of AplaCSP4 in the olfactory-driven behaviors of *A. planipennis*.

β-ionone is a common volatile compound in pine, cedar, and elm trees (*Henke et al., 2015*; *Zhang et al., 2022*), and it has been shown repellent activity against female *A. planipennis*, as well as *Phyllotreta cruciferae* and *Cnaphalocrocis medinalis* (*Cáceres et al., 2016*; *Sun et al., 2016*). Our results indicate that AplaCSP4 exhibits a high binding affinity for β-ionone. Although, the behavioral response of *A. planipennis* to ash tree volatiles such as dodecane, myrcene, ocimene, farnesene, (+)-limonene, and nerolidol (*Crook*

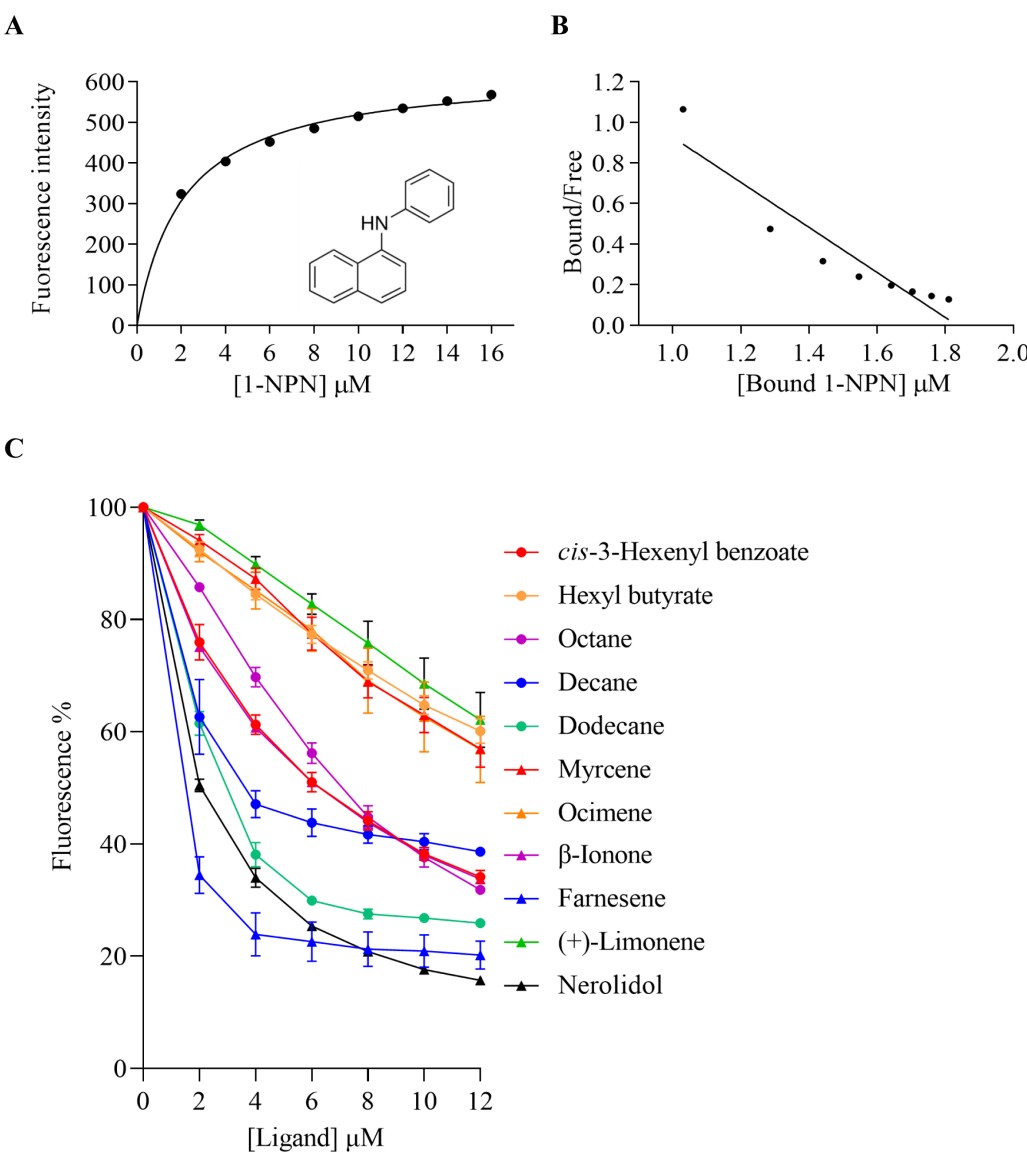

**Figure 4** **Binding properties of recombinant chemosensory protein 4 of *Agrilus planipennis* (AplaCSP4) to different ligand.** (A) presents the binding curve of the fluorescence probe 1-NPN with AplaCSP4. (B) shows the scatter plot of the bound/free ratio *versus* the concentration of bound 1-NPN. (C) indicates the competitive binding curves of volatile compounds to AplaCSP4.

*et al., 2008*; *Rigsby et al., 2017*; *Rodriguez-Saona et al., 2006*) have not been determined, the farnesene and nerolidol exhibited attractive activity to *Mythimna separata*, and the CSP14 of *M. separata* (MsepCSP14) played a critical role in the identification of farnesene and nerolidol (*Younas et al., 2022*). Our results demonstrate that AplaCSP4 has a particularly high binding affinity for farnesene and nerolidol. AplaCSP4 also exhibited binding capabilities to dodecane, myrcene, ocimene, and (+)-limonene. This suggests that AplaCSP4 may play a crucial role in the recognition of host plant volatile compounds,

**Table 1  Binding affinities of *Agrilus planipennis* chemosensory protein 4 (AplaCSP4) for 43 volatiles.**

| Ligands | Source | CAS number | Purity (%) | $K_D \pm$SE ($\mu$M) |
|---|---|---|---|---|
| *cis*-3-Hexenyl acetate[*] | TCI | 3681-71-8 | >97.0 | – |
| Methyl salicylate[*] | TCI | 119-36-8 | >99.0 | – |
| Hexyl acetate[*] | TCI | 142-92-7 | >99.0 | – |
| *trans*-2-Hexenyl acetate | TCI | 2497-18-9 | >97.0 | – |
| *cis-3*-Hexenyl benzoate | TCI | 25152-85-6 | >98.0 | 4.29 ± 0.18 |
| Hexyl butyrate | TCI | 2639-63-6 | >98.0 | 11.47 ± 0.42 |
| *cis*-3-Hexenyl isovalerate | TCI | 35154-45-1 | >98.0 | – |
| *cis*-3-Hexenyl isobutyrate | TCI | 41519-23-7 | >95.0 | – |
| Methyl benzoate | TCI | 93-58-3 | >99.0 | – |
| Nonanal | TCI | 124-19-6 | >95.0 | – |
| Benzaldehyde | TCI | 100-52-7 | >98.0 | – |
| Valeraldehyde | TCI | 110-62-3 | >95.0 | – |
| Decanal | TCI | 112-31-2 | >97.0 | – |
| Octanal | TCI | 124-13-0 | >98.0 | – |
| *trans*-2-Heptenal | TCI | 18829-55-5 | >95.0 | – |
| Hexanal[*] | TCI | 66-25-1 | >98.0 | – |
| *trans*-2-Hexenal[*] | TCI | 6728-26-3 | >97.0 | – |
| 1-Hexanol[*] | TCI | 111-27-3 | >98.0 | – |
| 1-Octanol | TCI | 111-87-5 | >99.0 | – |
| 1-Octen-3-ol | TCI | 3391-86-4 | >98.0 | – |
| *cis*-3-Hexen-1-ol[*] | TCI | 928-96-1 | >97.0 | – |
| *n*-Octane | TCI | 111-65-9 | >97.0 | 4.83 ± 0.11 |
| Decane | TCI | 124-18-5 | >99.5 | 2.87 ± 0.44 |
| Dodecane[*] | TCI | 112-40-3 | >99.5 | 1.93 ± 0.09 |
| Tetradecane | TCI | 629-59-4 | >99.5 | – |
| 4′-Ethylacetophenone[*] | TCI | 937-30-4 | >97.0 | – |
| 2-Hexanone | TCI | 591-78-6 | >98.0 | – |
| Isobornyl acetate | TCI | 125-12-2 | >90.0 | – |
| (±)-Citronellal | TCI | 106-23-0 | >98.0 | – |
| Myrcene[*] | TCI | 123-35-3 | >75.0 | 9.90 ± 0.34 |
| Ocimene[*] | SIGM | 13877-91-3 | ≥90 | 10.43 ± 1.11 |
| *β*-Ionone | TCI | 14901-07-6 | >95.0 | 4.23 ± 0.07 |
| (-)-*β*-Pinene[*] | TCI | 18172-67-3 | >94.0 | – |
| 1,8-Cineole[*] | TCI | 470-82-6 | >99.0 | – |
| Farnesene[*] | SIGM | 502-61-4 | >90 | 0.25 ± 0.07 |
| Citral | TCI | 5392-40-5 | >96.0 | – |
| (+)-Limonene[*] | TCI | 5989-27-5 | >99.0 | 11.34 ± 0.69 |
| (*R*)-(-)-Carvone | TCI | 6485-40-1 | >99.0 | – |
| Nerolidol[*] | TCI | 7212-44-4 | >97.0 | 1.38 ± 0.03 |

**Table 1** (*continued*)

| Ligands | Source | CAS number | Purity (%) | $K_D \pm SE$ ($\mu M$) |
|---|---|---|---|---|
| Linalool* | TCI | 78-70-6 | >96.0 | – |
| β-Caryophyllene* | TCI | 87-44-5 | >90.0 | – |
| Eugenol | TCI | 97-53-0 | >99.0 | – |
| Phenylacetonitrile | TCI | 140-29-4 | >98.0 | – |

**Notes.**

$K_D$, dissociation constant; $K_D < 1$, Strong binding; $1 \leq K_D \leq 10$, Moderate binding; $K_D > 10$, Weak binding; SE, Standard error.

We consider the AplaCSP4 had no binding with the tested ligands if the IC$_{50}$ values >16 μM and K$_D$ values were not to be calculated and are represented as ''–''. Data are means of three independent experiments and represents mean ± SE.

*denotes the volatiles found in the ash tree (*Rodriguez-Saona et al., 2006*; *Crook et al., 2008*; *Rigsby et al., 2017*).

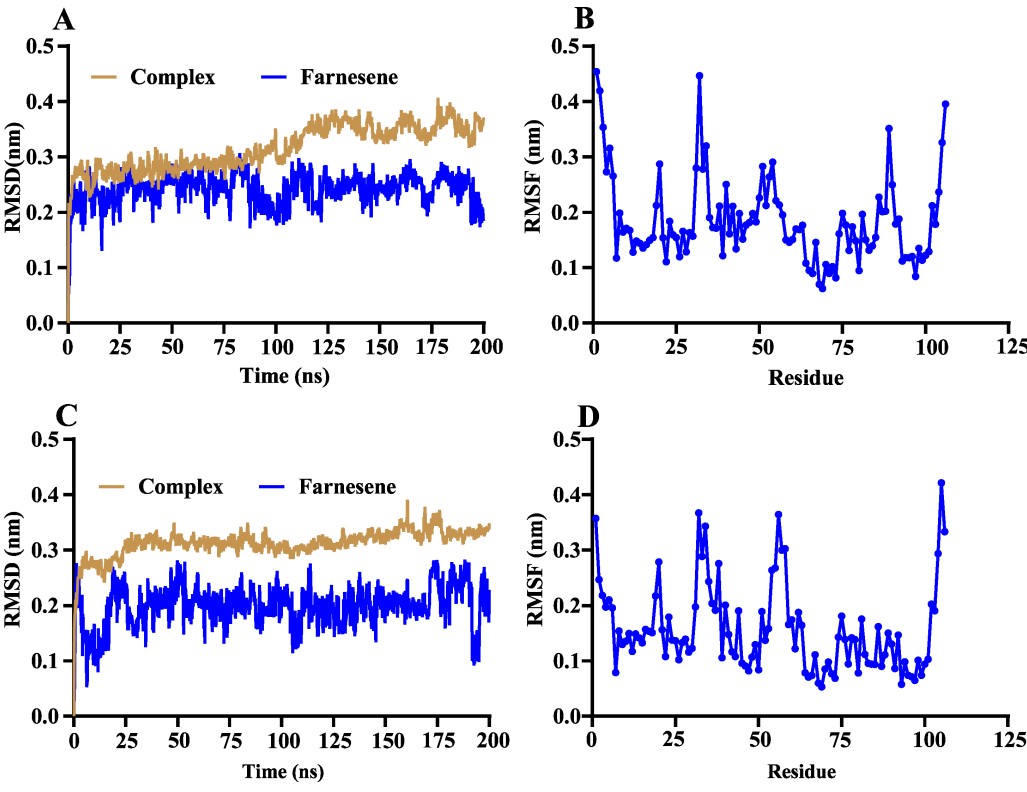

**Figure 5** **Root mean square deviation (RMSD) and root mean square fluctuation (RMSF) of *Agrilus planipennis* chemosensory protein 4 (AplaCSP4) and farnesene during two molecular dynamics simulations.** (A) and (B) indicate the RMSD and RMSF of the first molecular dynamics simulation, and C and D show the second molecular dynamics simulation.

while biological functional of AplaCSP4 in response the behavior for host plant volatile needed to further confirm.

To explore the binding mode of AplaCSP4 with its ligands, molecular docking and dynamics simulation were performed using farnesene, a volatile compound that exhibited the strongest binding affinity to AplaCSP4 in the fluorescence competition binding assay. The results demonstrated that farnesene binds within the same hydrophobic pocket of

**Table 2 Binding energy of *Agrilus planipennis* chemosensory protein 4 (AplaCSP4) with farnesene in dual-replicate dynamics simulations.**

| Repeat | dG (kJ/mol) | VDW (kJ/mol) | ELE (kJ/mol) | PB (kJ/mol) | SA (kJ/mol) | TdS |
|---|---|---|---|---|---|---|
| 1 | −133.176 ± 8.429 | −156.052 ± 8.319 | −1.821 ± 0.836 | 30.442 ± 3.987 | −21.999 ± 0.642 | −16.254 |
| 2 | −136.337 ± 8.413 | −154.460 ± 8.491 | −2.116 ± 1.168 | 26.659 ± 3.715 | −21.941 ± 0.601 | −15.521 |

**Notes.**

dG, Gibbs Free Energy Change; VDW, Van der Waals energy; ELE, Electrostatic energy; PB, Polar solvation energy; SA, Nonpolar solvation energy; Tds, Temperature times Entropy Change.

dG=VDW+ELE+PB+SA-Tds.

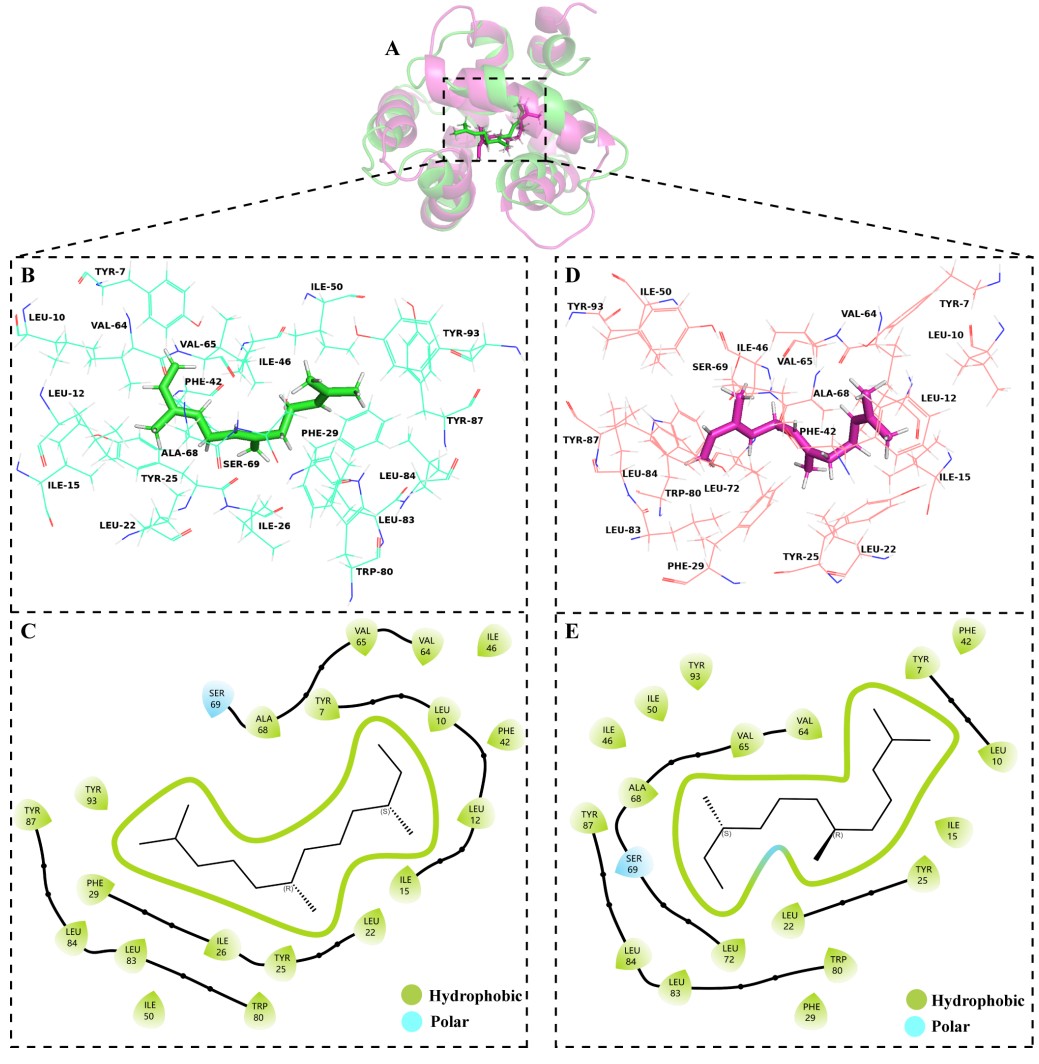

**Figure 6 Binding conformations of *Agrilus planipennis* chemosensory protein 4 (AplaCSP4) with farnesene in the last frame of dual-replicate molecular dynamics simulations.** (A) compares the binding sites of farnesene in AplaCSP4 from the two molecular dynamics simulations, with green representing the results of the first simulation and red representing the second. (B) and (C) show the binding pocket and interactions of farnesene with AplaCSP4 in the first molecular dynamics simulation, while (D) and (E) present the binding pocket and interactions of farnesene with AplaCSP4 in the second molecular dynamics simulation.

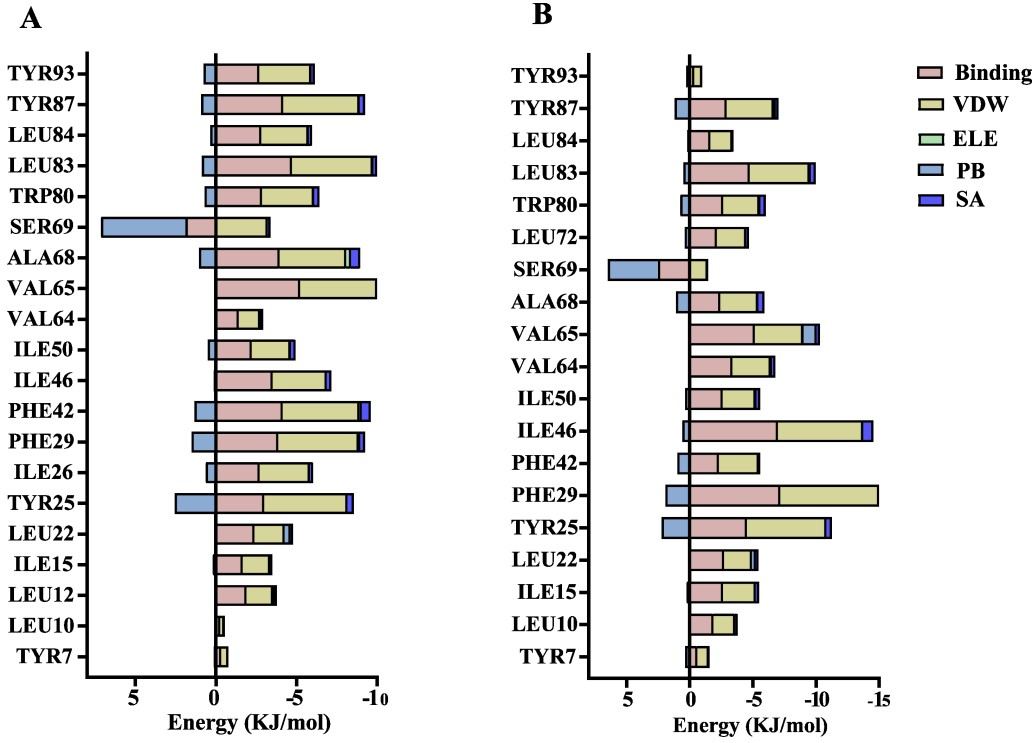

**Figure 7** **Contribution of key amino acids in the binding pocket of *Agrilus planipennis* chemosensory protein 4 (AplaCSP4) to binding energy.** (A) and (B) present the results of dual-replicate molecular dynamics simulations. VDW, Van der Waals energy; ELE, Electrostatic energy; PB, Polar solvation energy; SA, Nonpolar solvation energy; Binding = VDW+ELE+PB+SA.

AplaCSP4 in dual-replicate molecule dynamics simulations, and with strong binding energies of $-31.830 \pm 2.015$ kcal/mol and $-32.585 \pm 2.011$ kcal/mol, indicating that AplaCSP4 has a stable interaction with farnesene. Additionally, by analyzing the centroid distance between farnesene and the binding pocket of AplaCSP4, it was revealed that the binding mode of farnesene to AplaCSP4 remained relatively stable throughout the two molecular dynamics simulations, indicating that these residues (showing in Fig. 7) are relatively conserved and play a crucial role in stabilizing the ligand-binding pocket.

Our findings are consistent with previous studies on CSPs in other insects, which have shown that these proteins play a significant role in odorant detection and host recognition (*Li et al., 2021b*; *Yang et al., 2025*). The specific expression of AplaCSP4 in the antennae and its strong binding affinity for host volatiles highlight the importance of CSPs in the olfactory system of *A. planipennis*. The current research demonstrates only *in vitro* and *in silico* validation of AplaCSP4, but additional validations by *in vivo* experiments (gene silencing or editing) and behavioral assays are needed to demonstrate the actual biological function. This study extends our understanding of the molecular mechanisms underlying host recognition in invasive pests and may provide a foundation for developing novel pest management approaches.

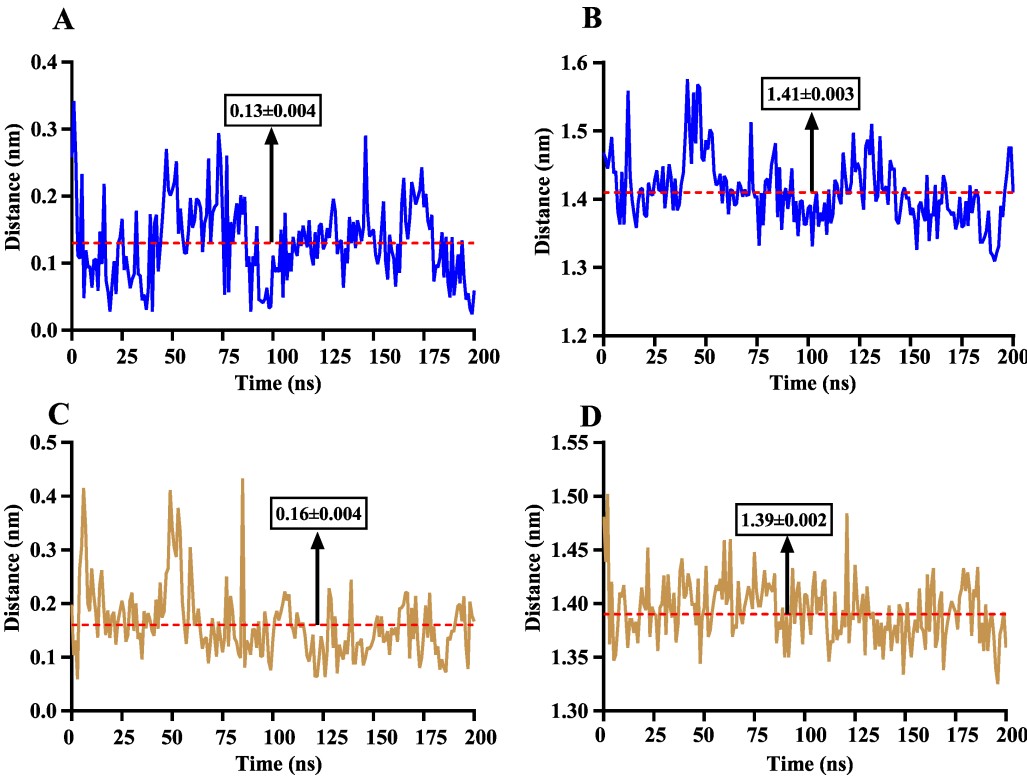

**Figure 8** The distance change between farnesene and the binding pocket of *Agrilus planipennis* chemosensory protein 4 (AplaCSP4) during two molecular dynamics simulations. (A) and (C) indicate the centroid distance between farnesene and key amino acids in the binding pocket, (B) and (D) present the centroid distance between the carbon atoms of farnesene and the side-chain carbon atoms of hydrophobic amino acids in the binding pocket. Blue represents the results of the first simulation, yellow represents the second simulation, and red dashed lines indicate the mean values, with numerical values display in black boxes as mean ± standard error.

## CONCLUSION

In conclusion, our study elucidates the role of AplaCSP4 in the olfactory perception of *A. planipennis* and its potential contribution to host recognition. The specific expression pattern, binding characteristics, and structural analysis of AplaCSP4 may provide valuable insights into the molecular mechanisms underlying plant volatiles identification in this invasive beetle. These findings have significant implications for developing novel olfactory-based pest management strategies that could help mitigate the ecological and economic damage caused by *A. planipennis*. Future research should focus on further characterizing the interactions between AplaCSP4 and host volatiles, as well as exploring the potential application of these findings in pest control programs.

## ACKNOWLEDGEMENTS

We would like to express gratitude to Zheng Kai for helping us collect the samples.

### Funding

This work was supported by the Promotion and Innovation Program of the Beijing Academy of Agriculture and Forestry Sciences (Grant No. KJCX20240403). The funders had no role in study design, data collection and analysis, decision to publish, or preparation of the manuscript.

### Grant Disclosures

The following grant information was disclosed by the authors:
Beijing Academy of Agriculture and Forestry Sciences: KJCX20240403.

### Competing Interests

The authors declare there are no competing interests.

### Author Contributions

- Ren Li conceived and designed the experiments, performed the experiments, analyzed the data, prepared figures and/or tables, authored or reviewed drafts of the article, and approved the final draft.
- Zehua Wang conceived and designed the experiments, performed the experiments, prepared figures and/or tables, authored or reviewed drafts of the article, and approved the final draft.
- Fan Yang performed the experiments, analyzed the data, prepared figures and/or tables, authored or reviewed drafts of the article, and approved the final draft.
- Guanghang Qiao performed the experiments, analyzed the data, prepared figures and/or tables, authored or reviewed drafts of the article, and approved the final draft.
- Jingjing Tu performed the experiments, analyzed the data, prepared figures and/or tables, authored or reviewed drafts of the article, and approved the final draft.
- Ang Sun conceived and designed the experiments, performed the experiments, analyzed the data, prepared figures and/or tables, authored or reviewed drafts of the article, and approved the final draft.
- Shanning Wang conceived and designed the experiments, analyzed the data, prepared figures and/or tables, authored or reviewed drafts of the article, and approved the final draft.

### DNA Deposition

The following information was supplied regarding the deposition of DNA sequences:
The sequences are available at NCBI: XM_018478165.1, XM_018479924.1, and XM_018476784.2.

### Data Availability

The raw data are available in the Supplementary Files.

## Supplemental Information

Supplemental information for this article can be found online at http://dx.doi.org/10.7717/peerj.19812#supplemental-information.

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
