# Peer review of "Functional characterization of antennae-enriched chemosensory protein 4 in emerald ash borer, Agrilus planipennis"

_PeerJ, doi:10.7717/peerj.19812_

## Round 0.1 · original submission · Major Revisions

Please consider the suggestions and corrections provided by the reviewers, and submit a revised version along with a rebuttal letter.

Reviewer 1 ·

Basic reporting

This is an interesting study reporting a CSP that is potentially involved in Agrilus planipennis interactions with its host plant volatiles. While CSPs detecting general odorants are well studied in insects, the manuscript demonstrates a key CSP capable of binding to host plant volatiles. However, the actual function of this CSP4 in A. planipennis is yet to be verified by in vivo experiments. Considering this, I request authors to limit their claims on AplaCSP4’s crucial role in host recognition as mentioned in the abstract. Rather I would recommend authors to claim only the actual experimental results and mention the importance of additional validations like in vivo experiments and behavioral assays to demonstrate the actual biological function.
Apart from that, I am glad to find that authors have done a good job by paying attention to detail in each step, following necessary technical standards in reporting the candidate protein. The introduction, methods, results and discussion are well written, but can be improved with minor additions. Figures are relevant but quality can be improved (Figure 4).
The Results could be elaborated with more details like selection of gene and minor addition of details. The discussion in written well, but could also be elaborated further. I recommend adding possible limitations like the reported CSP4 is one of the 14 CSPs in A. planipennis; which is capable of binding to selected volatiles tested in this study and has a binding pocket that strongly interacts with the particular ligand tested. Given this, it will be worth checking the docking of other potential ligands from the binding assay.

Experimental design

The methods use like antennal specific expression, quantification, followed by fluorescence competition binding assays and molecular docking and simulation provide enough data to 'predict the function' of the selected protein. However, the actual context in insect could be different and functional validation by gene knockdown and behavioural assys are essential for more claims on the function.
What I found missing was – how they selected AplaCSP4 for the analysis. As per Andersson et al., (2019) 14 CSPs have been reported in A. planipennis. I see that authors used tissue specific expression, but there is no data regarding other CSPs to support their claim! Is there any other antennal CSP? Adding supporting informaiton could solve this.
And if authors follow the naming of CSPs from Andersson et al., (2019) (recommended), did they check the expression pattern of AplaCSP12, a possible paralog of AplaCSP4 as per that paper? Please include selection of candidate gene in the experimental methods (appropriately in introduction as well).
Any validation regarding the candidacy of this protein compared to other CSPs will enhance this research. Further, are authors interested in the actual in vivo functional validation of AplaCSP4? Overall, it was an interesting paper to read and review. I hope authors can add minor details to make it more appealing.

Validity of the findings

The methodolgy used is sound and good; meeting necessary standards and reproducable. it is requested to recheck the high binding affinity. Necessary data are provided except the tissue speciifc expresison analysis and some technical details. The detials of missing information can be find at 'additional comments'. I am glad that, authors provided necessay supplementary data including the pdb files. The conclusion is short and clear.

Additional comments

Additional comments:
Title: It is appropriate to use the standard term: ‘chemosensory protein’ than chemoreception protein. Please follow this throughout the manuscript.
Is it antenna- specific or antennae-enriched?
Table 1: Please explain (*) used in caption.
L93: DNase treatment missing (for MIQE).
Please include tissue details like head/body was ‘with or without’ antennae/head? (Referring to Figure 2A, L:92-93 and L:117 for clarity).
L144: Please include primer efficiency. Also, in Table S1 (for MIQE).
L190: Please provide details of MD simulations and software used.
L213: Provide statistical significance (p value).
L218: Please add results here.
L243: The binding affinity seems to be very high. Please recheck your data and include additional details like binding site coordinates and final affinity preferably in kcal/mol. Did authors tried to compare other compounds from binding assay like Nerolidol?
L251: angstrom (Å) is the standard unit.
L251: Change ‘binding model’ to ‘interactions’ more appropriately.
L252: Please explain Figure 7 further in results.
L286-288 seems like results.
L291: these residues? Please specify if there are any particular residues in the results or else cite Figure 7.
Figure 4: Please add error bars and statistical information appropriately.
Figure 7: caption: correct ‘chemoreception Pprotein 4’ > chemosensory protein
Table S1: Please add table title, Tm and amplicon size
General: Add author contributions.

Reviewer 2 ·

Basic reporting

Authors Li et al. presented a study to functional characterize an antennae-specific chemoreception protein 4 (AplaCSP4) in emerald ash borer. They performed fluorescent binding assay with 1-NPN exhibited that AplaCSP4 has a broad binding spectrum to variety of volatiles from host plants and insects. Authors then chose the terpene, farnesene which exhibited the highest binding affinity in binding assay to conduct molecular docking and dynamics simulation analysis. Based on their data, authors concluded that “AplaCSP4 plays a crucial role in host recognition by binding to key plant volatiles, especially during the oviposition behavior of female adults when seeking suitable hosts.” This statement appears too strong given the current data (binding assay and molecular docking), suggest softening this claim. Following are some minor suggestions for improving the manuscript.

Abstract: Line 33, “suggests that AplaCSP4 MAY play a crucial role in the recognition of…” add a “may” into this sentence
M&M:
Line 165, Authors investigated 43 volatile organic compounds using a fluorescent binding assay. Please provide the rationale for selecting this specific set of volatiles.
Result:
Line 223, Add how many of these 11 compounds are found in the ash tree (Table 1) in the main text.
In Figure 5, the RMSD plot appears to stabilize after approximately 125 ns, which the authors interpret as the system reaching equilibrium. Please clarify how this conclusion was drawn — for example, by indicating the specific RMSD range used to define stability and whether both replicates showed consistent trends.
Discussion:
Line 282, “suggests that AplaCSP4 MAY play a crucial role in the recognition of…” add a “may” into this sentence, since further functional genomics or behavior assay are needed to confirm this role.

Experimental design

The experimental design is scientifically sound

Validity of the findings

The findings are generally supported by the presented data, including fluorescent binding assay, molecular docking and dynamics simulations. However, the conclusion that AplaCSP4 plays a crucial role in host recognition and oviposition behavior may be overstated, as no functional genomics or behavioral assays were conducted. The authors are encouraged to revise the language to reflect the predictive nature of their results and acknowledge the need for experimental validation.

Additional comments

This is a solid study, with a clear and well-organized presentation

---

## Round 0.2 · Minor Revisions

Please address the reviewer´s comments in a revised version.

Reviewer 1 ·

Basic reporting

Review of Li et. al.’s “revised manuscript” titled: Functional characterization of antennae-enriched chemosensory protein 4 in emerald ash borer, Agrilus planipennis

I am glad to find that authors have thoroughly revised the manuscript with recommended changes. The manuscript looks better with all sections significantly improved. I feel it is now clean for publication.
As I mentioned in the first review, my only additional recommendation is to stress on the need for in vivo experiments/behavioral assays to prove the actual biological (chemosensory) function of AplaCSP4. Since the study has validated it in heterologous in vitro system, I strongly recommend that authors make a note of that in the discussion.
In fact, it would be better if authors mention it in the abstract that in the current validation of AplaCSP4 is based on in vitro and in silico methods and not from in vivo studies.
I also agree with the authors’ responses to specific comments and concerns regarding the first version of this manuscript.

Experimental design

I agree with the changes made. the current experimental design meets the publication standards.

Validity of the findings

I agree with the changes made in the revised manuscript. The validity of the findings has been improved.

Additional comments

L27: change “The results indicated that” to ‘The predictions indicated that’
L33: change ‘plays’ to ‘play’
L386: change ‘olfactory perception’ to ‘odorant detection’.
L391: add a sentence, regarding the fact that, the current research demonstrates only in vitro and in silico validation of AplaCSP4, but additional validations by in vivo experiments (Gene silencing or editing) and behavioral assays are needed to demonstrate the actual biological function.
Being said that, I hope authors agree with the need for further research to validate the biological function.

Reviewer 2 ·

Basic reporting

The authors have addressed all my comments in their revision. I believe the manuscript is now suitable for publication.

Minor comment
Abstract
“AplaCSP4 may plays a crucial role…” change plays to play

Experimental design

no comment

Validity of the findings

no comment

Additional comments

no comment

---

## Round 0.3 · accepted · Accept

Thanks for addressing all the comments; your manuscript is now accepted in PeerJ.